# Pyrogallol-Phloroglucinol-6,6’-Bieckol from *Ecklonia cava* Improved Blood Circulation in Diet-Induced Obese and Diet-Induced Hypertension Mouse Models

**DOI:** 10.3390/md17050272

**Published:** 2019-05-08

**Authors:** Myeongjoo Son, Seyeon Oh, Hye Sun Lee, BoMi Ryu, Yunfei Jiang, Ji Tae Jang, You-Jin Jeon, Kyunghee Byun

**Affiliations:** 1Functional Cellular Networks Laboratory, Lee Gil Ya Cancer and Diabetes Institute, Gachon University, Incheon 21999, Korea; mjson@gachon.ac.kr (M.S.); seyeon8965@gachon.ac.kr (S.O.); hslee@gachon.ac.kr (H.S.L.); 2Department of Anatomy & Cell Biology, Graduate School of Medicine, Gachon University, Incheon 21936, Korea; 3Department of Marine Life Science, School of Marine Biomedical Sciences, Jeju National University, 1 Ara 1-dong, Jejudaehak-ro, Jeju 63243, Korea; bmryu@jejunu.ac.kr (B.R.); jiangyunfei0310@gmail.com (Y.J.); 4Aqua Green Technology Co., Ltd., Smart Bldg., Jeju Science Park, Cheomdan-ro, Jeju 63243, Korea; whiteyasi@gmail.com

**Keywords:** insufficient blood flow, *Ecklonia cava*, Pyrogallol-phloroglucinol-6,6’-bieckol, vascular dysfunction

## Abstract

Blood circulation disorders, such as hyperlipidemia and arteriosclerosis, are not easily cured by dietary supplements, but they can be mitigated. Although *Ecklonia cava* extract (ECE), as dietary supplements, are associated with improving the conditions, there are not many studies verifying the same. In this study, the beneficial effect of ECE and leaf of *Ginkgo biloba* extract (GBE), which is a well-known dietary supplement, were first confirmed in a diet induced-obese model. Afterwards, 4 phlorotannins were isolated from ECE, and their inhibitory effects on vascular cell dysfunction were validated. Pyrogallol-phloroglucinol-6,6-bieckol (PPB) was selected to be orally administered in two mice models: the diet induced obese model and diet induced hypertension model. After four weeks of administration, the blood pressure of all mice was measured, after which they were subsequently sacrificed. PPB was found to significantly improve blood circulation, including a reduction of adhesion molecule expression, endothelial cell (EC) death, excessive vascular smooth muscle cell (VSMC) proliferation and migration, blood pressure, and lipoprotein and cholesterol levels. Based on the excellent efficacy in diet-induced mouse models of obese and hypertension, our results demonstrate that PPB is a valuable active compound from among the phlorotannins that were isolated and it has the potential to be used in functional foods for improving the blood circulation.

## 1. Introduction

The blood circulatory system is responsible for supplying the body with oxygenated blood, and it promotes the delivery of hormones, nutrients, and medication to different organs. However, insufficient blood flow results in restricted delivery to some parts of the body, particularly the extremities [1]. Although not considered to be a disease *per se*, inadequate blood circulation causes health problems, such as atherosclerosis, coronary artery disease, and hypertension. 

The endothelium forms the active luminal surface of blood vessels and it constitutes an interface between circulating blood in lumen and vessel walls; the luminal surface is composed of a continuous single layer of endothelial cells (ECs) and it is supported by connective tissue. ECs control the vascular tone and contractile tension in walls, by releasing diverse relaxing and contracting factors, which include acetylcholine, ATP and ADP, substance P, bradykinin, histamine, thrombin, serotonin, reactive oxygen species (ROS), nitric oxide (NO), arachidonic acid and its metabolites, and vasoactive peptides [2]. Tunica media is distinguished from tunica intima by a transverse arrangement of its elastin and collagen fibers. The media layer is mainly composed of vascular smooth muscle cells (VSMCs) and its thickness is related to blood vessel type. VSMCs maintain homeostasis in the vascular system by actively contracting and relaxing, and the sympathetic nervous system, ROS, the renin-angiotensin-aldosterone system, and immune activation influence these functions [3]. Furthermore, EC and VSMC dysfunctions can dysregulate blood circulation and they are characterized by cell death, the over-expressions of cell adhesion molecules (e.g., E-selectin, intercellular adhesion molecule 1 (ICAM-1), and vascular cell adhesion molecule-1 (VCAM-1)) and Von Willebrand factor (VWF), platelet activation, excessive smooth muscle cell proliferation and migration, smooth muscle cell phenotypic modulation, and vascular permeability [4,5,6]. 

Recent times have seen a considerable interest in the use of functional foods to improve blood circulation, such as green tea, nuts, cayenne, *Ginkgo biloba* (GB), and *Ecklonia cava* (*E. cava*). GB is used as a herbal medicinal product to help in relieving peripheral arterial occlusive disease [7], and experimental evidence has shown that the leaf extract of GB (GBE) has numerous effects that are of medical interest, which include increasing the cerebral, peripheral, and microvascular blood flow and decreasing capillary permeability [8,9,10]. In addition, GBE is shown to protect against P-selectin-mediated immune cell adhesion and vascular inflammation [11]. The effects of GBE are attributed to the presence of two major fractions: flavonoids (under 24% of GBE) and ginkgolides (under 6% of GBE). It has been demonstrated that the ginkgolides reduce platelet aggregation and activation, and they act as platelet-activating factor antagonists, thereby encompassing the potential to improve blood circulation [12]. 

Of the various brown algal species, which include *E. cava*, *Sargassum coreanum*, *Sargassum thunbergii*, *Scytosiphon lomentaria*, *Ishige okamurae*, *Sargassum fulvellum*, and *Sargassum horneri*, it has been shown that *E. cava* has the potential to improve blood circulation, and its extracts are currently available as dietary supplements [13]. 

Coagulation studies have demonstrated that *E. cava* effectively prolonged thrombin time (TT), activated partial thromboplastin time (aPTT), and prothrombin time (PT), among brown algal [12]. In a Goldblatt mouse model of hypertension, increased systolic blood pressure and reduced blood flow in the renal arteries were significantly alleviated when compared to the values that were observed in control hypertension mice (15% reduction at 50 mg/kg *Ecklonia cava* extract (ECE)). In addition, the angiotensin converting enzyme (ACE) inhibitory effect, which is important for blood pressure control, was found to be most effective in the mice that were orally administrated 200 mg/kg ECE. The antihypertensive efficacy of ECE was confirmed by ACE inhibition [13]. Despite the well-known effect of ECE on blood circulation, the effects of the phlorotannins that comprise the ECE are not well known. Phlorotannins appear to be unique to marine algae and they are known to neutralize ROS activity; for example, they reduce the oxidizing capacity of oxidized low-density lipoprotein (LDL) and peroxynitrite, and it scavenges the 1,1-diphenyl-2-picrylhydrazyl (DPPH) radical [14]. 

Four active phlorotannins have been identified in *E. cava*, namely, dieckol (DK), 2,7-phloroglucinol-6,6-bieckol (PHB), pyrogallol-phloroglucinol-6,6-bieckol (PPB), and phlorofucofuroeckol A (PFFA) [15]. PPB treatment significantly modulates the monocyte migration by reducing the inflammatory factor levels and macrophage differentiation in vitro. In addition, PPB treatment markedly reduced monocyte-induced endothelial cell death and the related caspase levels, and reduces monocyte-induced vascular smooth muscle cell excessive proliferation and migration in vitro. Although PPB shown beneficial effects in vitro, there are not many studies regarding improving blood circulation in vivo. However, although PPB is shown to have beneficial effects in vitro, few studies have examined its effect on blood circulation in vivo. Therefore, the present study was undertaken to examine the effects of PPB on vascular dysfunction in obese and hypertension mouse models.

## 2. Results and Discussion 

### 2.1. ECE and GBE Treatments Improved ECs and VSMCs Functions In Vitro and In Vivo 

Palmitate-acid (PA) conjugated bovine serum albumin (BSA) treated cell assay was performed to confirm the efficacies of GBE and ECE. The EC survival assay result illustrated reduced EC survival ratios in the PA treated cells, which were dose-dependently higher in the PA plus ECE (PA/ECE) or GBE (PA/GBE) treated cells (Figure 1A). On the other hand, the VSMC proliferation and migration was inhibited after exposure to PA/ECE or PA/GBE as compared with PA treated cells (Figure 1B,C). The percentage of TUNEL positive cells were higher in diet induced obese mice that were orally administrated saline (DIO/saline) than in chow diet fed mice (Sham); these levels were significantly lower in ECE orally administrated obese mice (Obese/ECE) than in GBE orally administrated obese mice (Obese/GBE) (Figure 1D). In addition, the proliferating cell nuclear antigen (PCNA) positivity was less in ECE administrated obese mice when compared to saline administrated obese mice (Figure 1E) and, consequently, the media layers of blood vessels in hematoxylin and eosin stained aortas were thinner in ECE administrated mice than in the saline administrated mice (Appendix A). Furthermore, the levels of systolic, diastolic, and mean artery pressures (Figure 1F) were significantly lower in Obese/ECE mice than in Obese/Saline mice. These results indicate that, in the diet induced-obese mouse models, ECE and GBE were similarly effective in terms of improving the blood circulation.

### 2.2. PPB Effectively Reduced ECs and VSMCs Dysfunction In Vitro

In order to identify the active component of the ECE, four phlorotannins that were isolated from ECE were compared, as previously described [16]. The four isolated phlorotannins were confirmed to be DK, PHB, PPB, and PFFA. To investigate the protective effects of adhesion molecule expression on ECs as mouse endothelial cells, the ECs were exposed to either PBS, 0.25 mM of PA, or 0.25 mM of PA plus four phlorotannins for 48 h (Figure 2A). The expression levels of adhesion molecules, such as E-selectin, ICAM-1, VCAM-1, and vWF in PA/PBS treated SVEC4-10 were significantly higher than those in the PBS treated ECs, but the level in PA/PPB co-treated cells was lower than those in PA/PBS treated ECs (Figure 2B–E). Furthermore, survival ratios were significantly lower in the PA/PBS treated ECs than in the PBS treated ECs (Figure 2F). The survival ratios were significantly lower in PA/PBS treated SVEC4-10 cells than those in the PBS treated SVEC4-10 cells.

Apart from mouse endothelial cells, the exposure to PA/PBS also increased the excessive proliferation and migration of VSMCs. The treatment of VSMCs with PA resulted in significantly higher cell proliferating ratios than PBS treated cells; also, the proliferating ratios of PA/PPB co-treated MOVAS were significantly lower than PA/PBS treated cells (Figure 3A,B). Similarly, the migrating ratios were higher in PA/PBS treated VSMCs than PBS treated cells. The proliferating and migrating cell ratios of PA/PPB co-treated VSMCs were also lower than those of other phlorotannin treated cells (Figure 3C). 

### 2.3. PPB Helps Rescue Vascular Dysfunction in the Diet-Induced Mouse Models

When considering that PPB was more active than the other three active compounds in vitro, we examined the effects of PPB in diet-induced mouse models of obese and hypertension. Protein levels of the adhesion molecules, such as E-selectin, ICAM-1, VCAM-1, and vWF, were lower in the Obese/PPB mice group than in the Obese/Saline group. The adhesion molecule levels were also higher in the diet-induced hypertension mice model (Hypertension/Saline) than in the Sham group, but the expressions in Hypertension/PPB mice group were statistically significantly lower than in the Hypertension/Saline group (Figure 4A–F). 

The expressions of EC survival related molecules, such as PI3K-pAKT and pAMPK, in the Obese/PPB mice group and Hypertension/PPB mice group were significantly higher than those in the Obese/Saline mice group and Hypertension/Saline mice groups (Figure 5A–D). The TUNEL positive cells were also markedly lower in the Obese/PPB mice and Hypertension/PPB mice groups than in the Obese/Saline and Hypertension/Saline groups (Figure 5E,F). These results indicate that PPB modulates the expressions of adhesion molecules and cell survival in vivo.

In addition, the inhibitory effects of PPB on the excessive proliferation and migration of VSMCs were validated in the two animal models. The expression levels of pERK1/2 and pAKT in VSMCs were the highest in the Obese/Saline and Hypertension/Saline mice groups (Figure 6), which indicates the involvement in the observed reductions in numbers of proliferative VSMCs and in wall thicknesses. 

The PCNA positive cells were higher in the Obese/Saline and Hypertension/Saline mice groups than in the Sham group, but a reduced number of positive cells were observed in the Obese/PPB and Hypertension/PPB mice groups than in the Obese/Saline and Hypertension/Saline mice groups. Finally, aorta wall thickness was also significantly thinner in the Obese/PPB and Hypertension/PPB mice groups than in the Obese/Saline and Hypertension/Saline groups (Figure 7A–D). These results indicate that PPB inhibits excessive VSMC proliferation and migration, and it probably reduces blood vessel wall thickening in diet-induced mouse models of obese and hypertension. Some experiments further validated whether the uncontrolled-blood pressure and lipoproteins in these mouse models were improved after exposure to PPB. Systolic, diastolic, and mean blood pressures were higher in the Obese/Saline and Hypertension/Saline mice groups than in sham group, but the blood pressures were significantly lower in the Obese/PPB and Hypertension/PPB groups than in Obese/Saline and Hypertension/Saline mice groups (Figure 7E). In addition, serum total cholesterol, triglyceride, and LDL/HDL ratios were significantly lower in the Obese/PPB and Hypertension/PPB groups than in the Obese/Saline and Hypertension/Saline groups (Appendix A).

*E. cava* is a brown seaweed that is used in food products and it is the most common in the seas around Jeju Island [17]. Several studies have been conducted on the association between *Ecklonia cava* and cardiovascular health [13,14,15,17]. Some studies on phlorotannins that were isolated from ECE or ECE itself report potent anti-oxidant properties against toxin or stress induced ROS induction [18,19]. ECE is also reported to suppress lipopolysaccharide (LPS)-induced inflammatory responses in endothelial cells, and decreases the expressions of adhesion factors (E-selectin, VCAM-1, ICAM-1, and vWF) and immune cell adhesion in vitro. Additionally, ECE is reported to have anti-inflammatory and anti-oxidant effects in endothelial cells [20].

An interesting result of the present study is that the effect of GBE is not significantly different from that of GB, which is currently one of the 10 most popular supplements in the western world [7,8,9,10,11]. In rats, GBE supplementation for four weeks reduces the systolic, but not diastolic blood pressure, in a dose and time dependent manner [21,22]. Interestingly, systolic and diastolic blood pressures were also lower in Obese/ECE and Obese/GBE mice gropes than those in the Obese/Saline and there was no significant difference in between Obese/GBE and Obese/ECE mice group. However, mean artery blood pressure and blood vessel thicknesses were significantly lower in Obese/ECE than those in Obese/Saline (Figure 1F). 

*E. cava* is currently being explored for its health-promoting properties due to its high phlorotannin contents (~18% on a dry weight basis). It is known to contain numerous phlorotannins, which includes dieckol, PFFA, PHB, PPB, phlorofucofuroeckol, 6,6′-bieckol, 8,8′-bieckol, 2-*O*-(2,4,6-trihydroxyphenyl)-6,6′-bieckol, eckstolonol, 2-phloroeckol, 1-(3′,5′-dihydroxyphenoxy)-7-(2″,4″,6-trihydroxyphenoxy)-2,4,9-trihydroxydibenzo-1,4-dioxin, eckol, 7-phloroeckol, phlorotannin A, and phloroglucinol [23,24,25,26,27,28]. The phlorotannins have a resilient dibenzo-1,4-obesityxin backbone, which presumably aids molecular interactions with various biomolecules [29,30]. Interestingly, the anti-oxidant effects of these phlorotannins are related to the number of hydroxyl groups that are present. According to a study by Li et al., dieckol and 6,6′-bieckol, which both containing more than 10 hydroxyl groups, have a higher anti-oxidant efficacy than phloroglucinol or eckol (less than 10 hydroxyl groups) [26]. Thus, since PPB possesses 15 hydroxyl groups, it might also be expected to improve blood circulation effects, such as monocyte migration and macrophage polarization.

We isolated four phlorotannins from ECE and observed that all of the reduced adhesion molecules expression and increased EC survival. In addition, PPB effectively increases the EC survival and decreases the adhesion molecule expression in diet-induced mouse models of obese and hypertension (Figure 4). Previous studies have shown that adhesion molecule expression and pAKT-PI3K negatively affect EC survival [15,31]. The effects of PPB on VSMCs, which play important roles in blood circulation-related diseases, were also examined. Our results show that exposure to PA causes VSMC dysfunction, and PPB treatment helps to reduce the excessive proliferation in MOVAS and in these two mouse models.

High fat and high cholesterol diets induce the de-differentiation of contractile vascular smooth muscle cells into synthetic smooth muscles cells, and these phenotype changes play important roles in vascular diseases, like atherosclerosis [32]. Synthetic cells exhibit excessive proliferation, migration, and neo-intima formation. Proliferative cell (marked PCNA) and intima-media ratio were found to be significantly lower in the Obese/PPB and Hypertension/PPB group than those in the Obese/Saline and Hypertension/Saline group (Figure 5). Interestingly, PPB as an active phlorotannin from ECE significantly reduces the levels of lipoprotein, cholesterols, and blood pressure (Figure 7 and Appendix A). This potency is presumably related to the anti-oxidant efficacy of PPB and the number of hydroxyl groups that are attached to its ring system [27,33].

## 3. Materials and Methods 

### 3.1. Preparations of ECE and GBE and Isolation of Four Phlorotannins

ECE was obtained from Aqua Green Technology Co. Ltd. (Jeju, Korea). Briefly, *Ecklonia cava* were washed, air-dried for 48 h at room temperature, ground, and extracted while using 50% aqueous ethanol for 12 h at 85 °C. The mixture was then filtered, concentrated, sterilized by heating for 60 min. at 85 °C, and spray-dried. The four phlorotannins that were used in this study were isolated from ECE using a previously described method [16]. Briefly, centrifugal partition chromatography (CPC) was performed while using a two-phase solvent system comprised of water/ethyl acetate/methanol/n-hexane (7:7:3:2, *v/v/v/v*). For separation, the stationary phase was an organic solution that was filled in the CPC column; the mobile phase was pumped into the column in a descending mode at the flow rate 2 mL/min. The positive control GBE was purchased ready to use from Hanzhong TRG Biotech Co. Ltd. (Hanzhong, China).

### 3.2. Endothelial Cell Survival Assay

SVEC4-10 cells (10^4^) were purchased from the Korean cell line bank (Seoul, Korea). The cells were seeded in wells of 96-well plates and, after 24 h incubation, the attached cells were rinsed twice with PBS. The cells were then treated with BSA plus PBS (PBS), PA plus PBS (PA/PBS), or PA plus phlorotannin (PA/DK, PA/PHB, PA/PPB, or PA/PFFA) for 48 h. The treated cells were rinsed twice with PBS and cell survival was determined using the Transdetect cell counting kit (Transgen Biotech Co., Ltd., Beijing, China). Absorbance was measured at 450 nm using a microplate reader (Spectra max plus, Molecular Devices, San Jose, CA, USA).

### 3.3. VSMC Proliferation Assay

MOVAS cells (10^4^) were purchased from the Korean cell line bank (Seoul, Korea). The cells were seeded in 96-well plates; after 24 h incubation, the attached cells were rinsed twice with PBS and subsequently treated with BSA plus PBS (PBS), PA plus PBS (PA/PBS), or PA plus phlorotannin (PA/DK, PA/PHB, PA/PPB, or PA/PFFA) for 4 h, rinsed twice with PBS, and cell proliferation assay was determined while using the Transdetect cell counting kit. Absorbance was measured at 450 nm using a microplate reader (Spectra max plus, Molecular Devices).

### 3.4. VSMC Trans-Well Migration Assay

MOVAS cells (5 × 10^5^) were seeded in the upper compartments of trans-well chambers and they were treated with BSA plus PBS (PBS), PA plus PBS (PA/PBS), or PA plus phlorotannin (PA/DK, PA/PHB, PA/PPB, or PA/PFFA) for 24 h. The attached cells were rinsed twice with PBS, and cells remaining in the upper compartments of the trans-well chamber were thoroughly removed with cotton swabs. The migrating cells were detected using the Transdetect cell counting kit to quantify cell migration to lower compartments; absorbance was measured at 450 nm using a micro plate reader (Spectra max plus).

### 3.5. Quantitative Real Time Polymerase Chain Reaction (qRT-PCR)

Cellular RNA was isolated using RNAiso Plus (TAKARA, Kusatsu, Japan), according to the manufacturer’s instructions. Briefly, the cell pellets were resuspended in 1 mL of RNAiso Plus by pipetting, mixed with 0.1 mL chloroform (Amresco, Cleveland, OH, USA), and centrifuged at 12,000× *g* for 15 min at 4 °C. The supernatants were mixed with 0.25 mL 100% isopropanol, and isolated RNA pellets were washed with 70% ethanol and centrifuged at 7500× *g* for 5 min. Dried pellets were dissolved in 30 µL diethyl pyrocarbonate (DEPC) water, and RNA was quantified while using a Nanodrop 2000 (Thermo Fisher Scientific, Waltham, MA, USA). cDNA was prepared from RNA using a cDNA synthesis kit (PrimeScript™, TAKARA). qRT-PCR was performed to determine the RNA levels. The primer was mixed with distilled water and then placed in 384-wells. Template (cDNA) and SYBR green (TAKARA) were subsequently added, and then validated using PCR machine (Bio-Rad, Hercules, CA, USA). Appendix A lists the genes of interest.

### 3.6. Diet Induced Mice Model

C57BL/6N male mice (seven weeks old) were obtained from Orient Bio (Sungnam, Korea) and maintained under controlled conditions (room temperature 23 °C and 50% humidity under a 12 h dark/light cycle). 

High fat diet-induced obese mice model (obese) was used in this study. Briefly, the mice were randomly assigned to five groups: (group 1, Sham) the mice were fed a regular chow diet (PicoLab Rodent diet, Fort Worth, TX, USA) for four weeks and then orally administrated 0.9% normal saline for four weeks; (group 2, Obese/Saline) mice were fed a 45% high fat diet (HFD; Research Diet Inc., New Brunswick, NJ, USA) for four weeks to procure diet-induced obese mice, and then orally administrated saline for four weeks; (group 3, Obese/ECE) mice were fed a 45% HFD for four weeks and then orally administrated ECE (70 mg/kg/day) for four weeks; (group 4, Obese/GBE) mice were fed a 45% HFD for four weeks and then orally administrated GBE (500 mg/kg/day) for four weeks [34]. To confirm the effects of Pyrogallol-Phloroglucinol-6,6-Bieckol (PPB) as a single phlorotannin from ECE, (group 5, Obese/PPB) the mice were fed HFD for four weeks and then orally administrated PPB (2.5 mg/kg/day (ECE contains 3.57% PPB)) for four weeks.

Hypertension mice model (Hypertension) was used in this study. Briefly, the mice were randomly assigned to two groups: (group 1, Hypertension/Saline) mice were fed with high cholesterol diet (Harlan Laboratories, Indianapolis, IN, USA) with salt (0.03 mg/kg/day, Hanju Corp., Ulsan, Korea) for four weeks and then orally administrated 0.9% saline for four weeks; (group 2, Hypertension/PPB) mice were fed high cholesterol diet with salt for four weeks and then orally administrated PPB (2.5 mg/kg/day) for four weeks. At eight weeks after treatment commencement, all of the mice were sacrificed in accordance with the ethical principles issued by the Institutional Animal Care and Use Committee of Gachon University (approval number LCDI-2017-0034). Appendix A contain the schemes for the entire constructed animal model.

### 3.7. DAB Immunohistochemistry

Aorta paraffin block tissues were sectioned at 7 µm, placed on slides, and dried at 37 °C for 24 h. The slides with the paraffin embedded sections were incubated in 0.3% H_2_O_2_ (Sigma-Aldrich, saint louis, MO, USA) for 30 min, rinsed three times with PBS, incubated in animal serum to block antibody binding, incubated with primary antibodies (listed in Appendix A), and then rinsed three times with PBS. The probed slides were then exposed to biotinylated secondary antibodies from the ABC kit (Vector Laboratories, Burlingame, CA, USA), incubated for 1 h in blocking solution, and rinsed three times with PBS. Thr slides were developed with DAB (3,3-diaminobenzidine) substrate for 5 to 15 min, mounted with cover slips and DPX mounting solution (Sigma-Aldrich), and visualized by light microscopy (Olympus Optical Co., Tokyo, Japan). Appendix A lists the antibodies used this study.

### 3.8. Immunofluorescence (IF)

Paraffin block sections (7 µm) were de-paraffinized, incubated in animal serum to block antibody binding, incubated with primary antibodies (listed in Appendix A) for two days at 4 °C, rinsed tree times with PBS, incubated for 1 h with fluorescence conjugated secondary antibody, and then rinsed three times with PBS. The rinsed sections were then incubated for 5 min. in DAPI solution, rinsed three times with PBS, and mounted with cover slip using vector shield solution (Vector Laboratories). Fluorescence was detected using a confocal microscope (LSM 710, Carl Zeiss, Oberkochen, Germany). Appendix A lists the antibodies used this study.

### 3.9. Blood Pressure Measurements

Blood Pressure is measured using by a noninvasive tail-cuff CODA system (Kent Scientific Corp., Torrington, CT, USA), as previously described in mice [35]. The mice sublimation was conducted for 5 min. for two days, 10 min. for two days, and 15 min. for two days in each group (total six days), and the last seven days we measured blood pressure.

### 3.10. Data Analysis

The results are expressed as the means ± standard deviations (S.D.). Kruskal–Wallis tests were used to determine the differences between groups, and applying the Mann–Whitney U test in SPSS ver. 22 software completed post-hoc comparisons. Significant differences are indicated, as follows: asterisk (*) versus PBS and Sham; $ versus PA/PBS, Obese/Saline and Hypertension/Saline; and, # versus PA/PPB, Obese/PPB, and Hypertension/PPB.

## 4. Conclusions

The efficacy of ECE for improving blood circulation was confirmed by administration to a high fat diet-induced mouse model for four weeks. ECE contains phlorotannins, such as dieckol (DK), 2,7-phloroglucinol-6,6-bieckol (PHB), pyrogallol-phloroglucinol-6,6-bieckol (PPB), and phlorofucofuroeckol A (PFFA). Especially, PPB markedly reduces adhesion molecule expression, EC death and excessive migration, and the proliferation of VSMCs in vitro and in the obese and hypertension mouse models. Additionally, PPB, as a dietary supplement, remarkably reduces the blood pressure and the serum lipoprotein levels in vivo. As expected, PPB plays an important role as the active substance in ECE for improving blood circulation. Currently, it is anticipated that functional foods, such as substances that are effective in alleviating various blood circulation related problems, will be effective in improving blood circulation. It is believed that such substances have the potential for future applications in the clinical setting.

## Figures and Tables

**Figure 1 marinedrugs-17-00272-f001:**
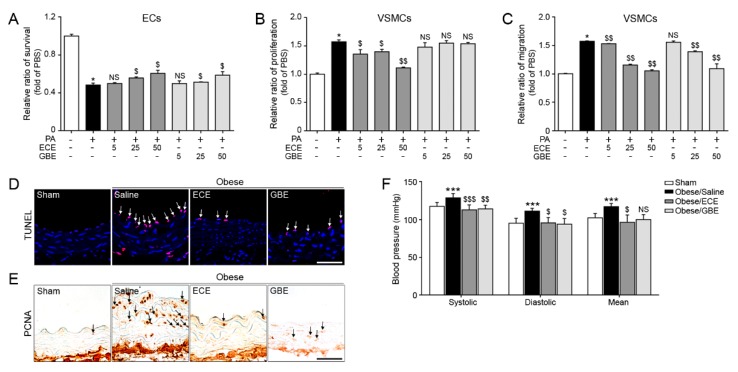
Effects of *Ginkgo biloba* extract (GBE) and *Ecklonia cava* extract (ECE) on vascular dysfunction in vitro and in vivo (**A**) endothelial cells (ECs) (SVEC4-10; mouse endothelial cells) were exposed to Palmitate/Phosphate-buffered saline (PA/PBS), Palmitate/ECE (PA/ECE) or Palmitate/GBE (PA/GBE). The survival ratios of ECs were measured using a cell survival assay. (**B**,**C**) PA/PBS, PA/ECE, or PA/GBE were administered to vascular smooth muscle cells (VSMCs) (MOVAS, mouse vascular smooth muscle cells), and proliferating to migrating VSMC ratios were measured by respective assays. (**D**) Confocal microscopic images showing apoptotic cells as TUNEL positive cells (red, arrows) and DAPI stained nuclei (blue). Scale bar = 50 μm (**E**) Light microscopic images showing proliferating cell nuclear antigen (PCNA) positive cells (brown, arrow) in obese mice. Scale bar = 10 μm (**F**) Blood pressure plots of systolic, diastolic, and mean artery blood pressure in obese mice. *, P < 0.05 and ***, P < 0.001 vs. the PBS (Sham) group; $, P < 0.05, $$, P < 0.01 and $$$, P < 0.001 vs. PA (or DIO/Saline) group; NS, not significant. Results are presented as means ± SD. ECE, *E. cava* extract; GBE, leaf of GB extract; PCNA, Proliferating cell nuclear antigen; TUNEL, Terminal deoxynucleotidyl transferase dUTP nick end labeling.

**Figure 2 marinedrugs-17-00272-f002:**
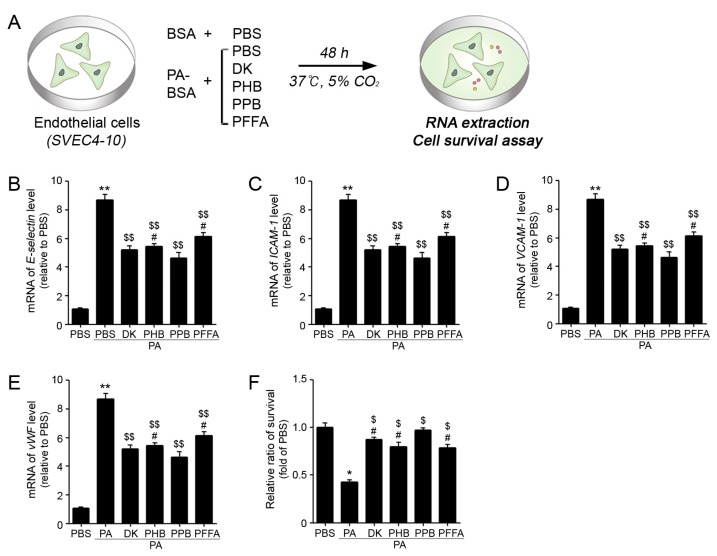
Comparison of the four phlorotannins isolated from ECE on EC adhesion molecule expression and cell survival in vitro. (**A**) Illustration explains palmitate induced EC dysfunction model in vitro. (**B**–**E**) mRNA expression levels of the adhesion molecules, including E-selsectin, ICAM-1, VCAM-1 and vWF in ECs as determined by qRT-PCR. (**F**) The graph shows PA with or without phlorotannin (dieckol (DK), 2,7-phloroglucinol-6,6-bieckol (PHB), Pyrogallol-Phloroglucinol-6,6-Bieckol (PPB), and phlorofucofuroeckol A (PFFA)) treated EC survival ratio. *, P < 0.05 and **, P < 0.01 vs. the PBS group; $, P < 0.05 and $$, P < 0.01 vs. PA/PBS group; #, P < 0.05 vs. PA/PPB. PA, palmitate conjugation to BSA; DK, dieckol; PHB, 2,7-phloroglucinol-6,6-bieckol; PPB, pyrogallol-phloroglucinol-6,6-bieckol; PFFA, phlorofucofuroeckol A.

**Figure 3 marinedrugs-17-00272-f003:**
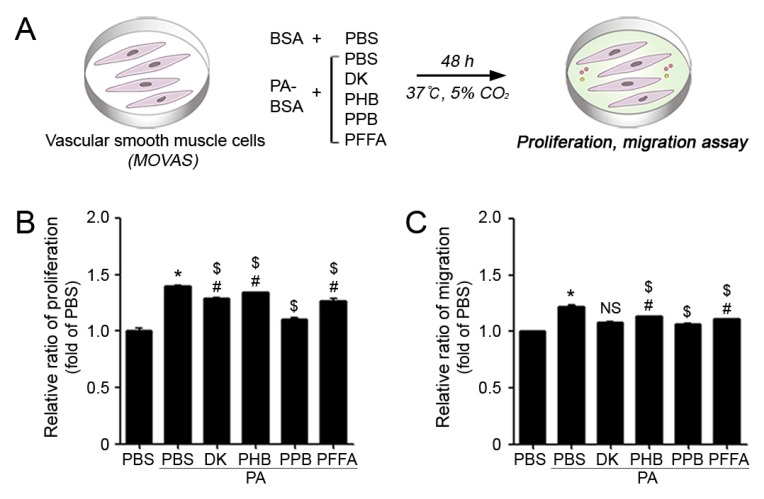
Comparison of the four phlorotannins isolated from ECE on excessive proliferation and migration of VSMCs in vitro. (**A**) Illustration explains palmitate conjugation to BSA (PA) induced VSMC dysfunction model in vitro. (**B**) PA with or without phlorotannin treated VSMCs were measured using a proliferation assay, and (**C**) trans-well migration assay. *, P < 0.05 vs. the PBS group; $, P < 0.05 vs. PA group; and, #, P < 0.05 vs. the PA/PPB group. NS, not significant. PA, palmitate conjugation to BSA; DK, dieckol; PHB, 2,7-phloroglucinol-6,6-bieckol; PPB, pyrogallol-phloroglucinol-6,6-bieckol; PFFA, phlorofucofuroeckol A.

**Figure 4 marinedrugs-17-00272-f004:**
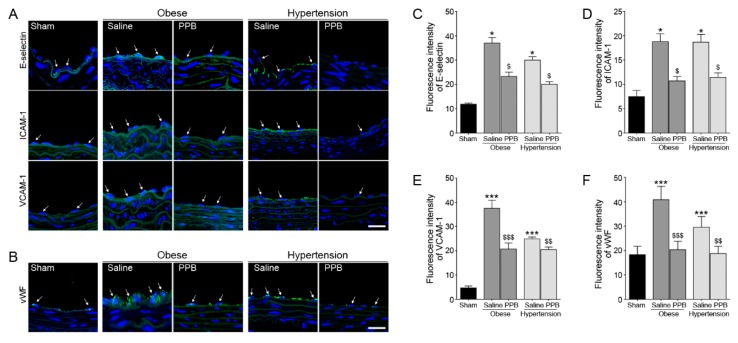
Inhibitory effects of the PPB isolated from ECE on EC adhesion molecule expression in diet-induced mouse models of obese and hypertension. (**A**,**B**) Arrows in confocal microscopic images show adhesion molecule expressions (green, arrow; E-selectin, ICAM-1, VCAM-1, vWF) or nuclei (DAPI, blue) in obese and hypertension mouse models. (**C**–**F**) Quantitative graphs show fluorescence intensity from representative images (dark gray bar, obese mouse model; gray bar, hypertension mouse model). Scale bar = 50 μm *, P < 0.05 and ***, P < 0.001 vs. the Sham group; $, P < 0.05, $$, P < 0.01, and $$$, P < 0.001 vs. Obese/Saline or Hypertension/Saline group.

**Figure 5 marinedrugs-17-00272-f005:**
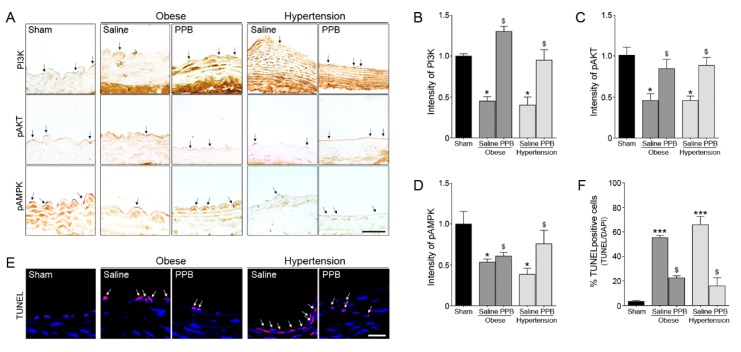
Inhibitory effects of the PPB isolated from ECE on EC dysfunction in diet-induced mouse models of obese and hypertension. (**A**) Light microscopic images showing the EC survival related molecules, such as PI3K, pAKT and pAMPK (brown, arrow). (**B**–**D**) Quantitative graphs show intensity from representative images (dark gray bar, obese mouse model; gray bar, hypertension mouse model). Scale bar = 10 μm. (**E**,**F**) TUNEL positive cells (red, arrow) indicate apoptotic cells, and nuclei stained DAPI (blue). Quantitative graphs show percentage of TUNEL positive cells from representative images. Scale bar = 50 μm. *, P < 0.05 and ***, P < 0.001 vs. the sham group; $, P < 0.05 vs. Obese/Saline or Hypertension/Saline group. PPB, pyrogallol-phloroglucinol-6,6-bieckol.

**Figure 6 marinedrugs-17-00272-f006:**
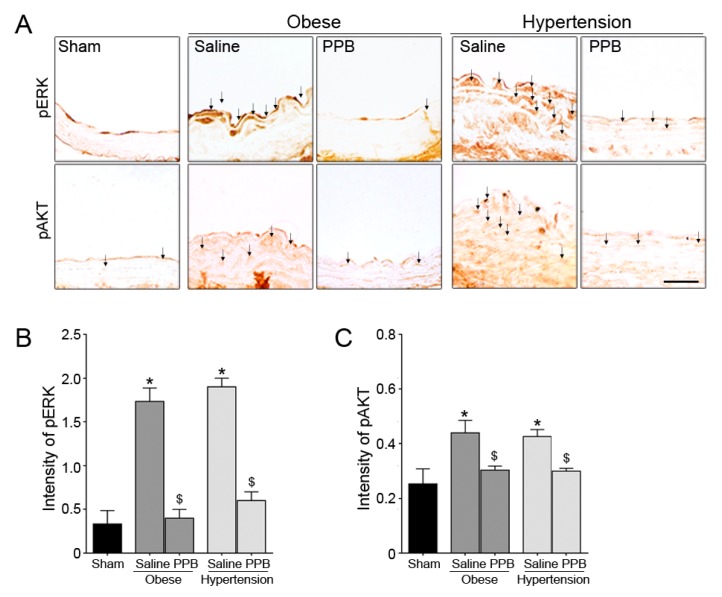
Inhibitory effects of the PPB isolated from ECE on VSMC dysfunction related molecules in diet-induced mouse models of obese and hypertension. (**A**) Light microscopic images showing abnormally high expressions of proliferation and migration related molecules (pERK1/2 and pAKT) in VSMCs in the Obese and in Hypertension models. (**B**,**C**) Quantitative graphs show intensity from representative images (dark gray bar, Obese mouse model; gray bar, Hypertension mouse model). Scale bar = 10 μm. *, P < 0.05 vs. the sham group; $, P < 0.05 vs. Obese/Saline or Hypertension/Saline group. PPB, pyrogallol-phloroglucinol-6,6-bieckol.

**Figure 7 marinedrugs-17-00272-f007:**
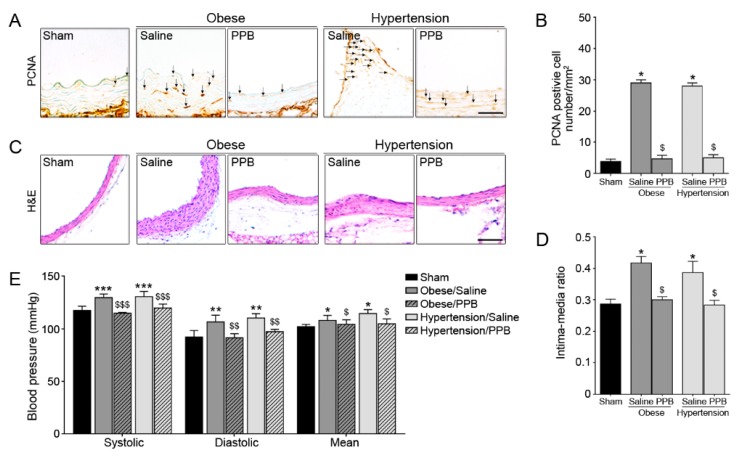
Effects of PPB isolated from ECE on vascular dysfunction and blood pressure in diet-induced mouse models of obese and hypertension (**A**,**C**) Light microscopic images showing proliferation marker (PCNA) and H&E stained blood vessels. (**B**,**D**) Plots showing numbers of PCNA positive cells and intima-media ratios obtained using representative H&E images (dark gray bar, Obese mouse model; gary bar, Hypertension mouse model). (**E**) Blood pressures, such as systolic, diastolic, and mean artery pressure in Obese and Hypertension mouse models. Scale bar = 10 μm (PCNA) and 200 μm (H&E). *, P < 0.05 vs. Sham group; $, P < 0.05 vs. Obese/Saline or Hypertension/Saline mice group. PCNA, Proliferating cell nuclear antigen; H&E, Hematoxylin and eosin. *, P < 0.05, **, P < 0.01 and ***, P < 0.001 vs. the sham group; $, P < 0.05, $$, P < 0.01 and $$$, P < 0.001 vs. Obese/Saline or Hypertension/Saline group. PPB, pyrogallol-phloroglucinol-6,6-bieckol.

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
