# Peer review of "Pyrogallol-Phloroglucinol-6,6’-Bieckol from Ecklonia cava Improved Blood Circulation in Diet-Induced Obese and Diet-Induced Hypertension Mouse Models"

_marinedrugs, 2019, doi:10.3390/md17050272_

Reviewer 1 Report

The study conducted by about Son et al investigated the effects of Pyrogallol-phloroglucinol-6,6'-bieckol from Ecklonia cava on blood circulation in diet-induced obesity and diet-induced hypertension mouse models. In fact the methodology and the experimental parts are conducted well but this manuscript needs major revision to reach the high level of Marin Drugs journal and my comments are:

1.     Shortening the background section of the Abstract (Too long).

2.     Avoid the use of (We or Our) throughout the whole manuscript.

3.     The plants Latin scientific names must be in Italic manner.

4.     Too many abbreviations throughout the whole manuscript try to reduce them to be this manuscript more readable for other none medical professions.

5.     Reference number 2 and many other references are too old and try to find other updated ones.

6.     Many sentences throughout the manuscript needs to be rephrased.

7.     Figure 1 4 and 5 , need to be divided into two figures (too many images in one figure).

8.      Very short conclusion? you can add some recommendations.

9.     Many correction required to be established (grammatical and editing) and needs to be checked by native speaker or by any editing proof services.

10.  Other required corrections you kindly can find in the attached pdf file.

Best Wishes

Author Response

Response to Reviewer 1 Comments

The study conducted by about Son et al investigated the effects of Pyrogallol-phloroglucinol-6,6'-bieckol from Ecklonia cava on blood circulation in diet-induced obesity and diet-induced hypertension mouse models. In fact, the methodology and the experimental parts are conducted well but this manuscript needs major revision to reach the high level of Marin Drugs journal and my comments are:

Point 1: Shortening the background section of the Abstract (Too long).

Response 1: We appreciate this comment. As per the reviewer’s suggestion, the background section of the abstract has been shortened, which we think will help readers understand it easily. The original abstract contained 643 characters (including spaces), whereas the revised background section has 291 characters. We have replaced with the modified background section of abstract. As well as background section, the whole abstract was modified to reflect the modified result.

Original abstract (background section)

Blood   circulation disorders can cause various diseases including hyperlipidemia,   arteriosclerosis, myocardial infarction, and stroke, and as a result a   variety of diets or dietary supplements have been developed to alleviate   circulatory disorders. Functional food supplements such as Ginkgo biloba   (GB), Ecklonia cava extract (ECE)   are often used in traditional medicine and GB extract (GBE) are associated   with improving blood circulation. Especially ECE contains a lot of   phlorotannins and has well known anti-oxidative properties, but few studies   have been undertaken to identify its active components for improvement of   blood circulation. In this study, we first compared the ECE with the GBE to   know if the ECE is as efficacious as the well-known GBE in diet induced obese   mice model. next, we isolated single phlorotannins of ECE using CPC and HPLC   and confirmed inhibitory effects of endothelial cell and vascular smooth   muscle cell dysfunction. Among phlorotannins such as dieckol (DK),   2,7-phloroglucinol-6,6-bieckol (PHB), pyrogallol-phloroglucinol-6,6-bieckol   (PPB) and phlorofucofuroeckol A (PFFA), PPB markedly reduced adhesion   molecule expression, cell death on ECs and excessive migration, proliferation   on VSMCs in vitro. PPB was selected, and orally administered to two mice   models; diet induced obese and diet induced hypertension model. After 4 weeks   of treatment, all mice measured blood pressure and sacrificed. PPB was found   to significantly reduce blood pressure, lipoprotein and cholesterol levels   and improve blood circulation in mice models of obesity and hypertension. We   selected PPB as an active compound among the phlorotannins, which is an   important compound in functional food for improving blood circulation because   it shows excellent effect in diet-induced mouse models of obesity and   hypertension.

Modified abstract (background section)

Blood circulation disorders such   as hyperlipidemia and arteriosclerosis are not easily cured by dietary   supplements, but can be mitigated. Although Ecklonia cava extract (ECE) as dietary supplements are associated   with improving the conditions, there are not many studies verifying the same. In this   study, the beneficial effect of ECE and leaf of Ginkgo biloba extract (GBE), a   well-known dietary supplement, were first confirmed in a diet induced-obese   model. Next, 4 phlorotannins were isolated from ECE, and their inhibitory   effects on vascular cell dysfunction were validated. PPB was selected to be orally   administered in two mice models: the diet induced obese model and diet   induced hypertension model. After 4 weeks of administration, blood pressure of   all mice were measured, after which they were subsequently sacrificed. PPB   was found to significantly improve blood circulation, including reduction of   adhesion molecule expression, EC death, excessive VSMC proliferation and   migration, blood pressure, and lipoprotein and cholesterol levels. Based on   the excellent efficacy in diet-induced mouse models of obesity and   hypertension, our results demonstrate that PPB is a valuable active compound from   among the phlorotannins isolated, and has the potential to be used in   functional foods for improving the blood circulation.

Point 2: Avoid the use of (We or Our) throughout the whole manuscript.

Response 2: We appreciate your important comments. Accordingly, ‘we’ or ‘our’ word have been replaced by ‘this’ or ‘it’ throughout the manuscript or sentences were rephrased. The changed examples are summarized below.

1.        We previously reported on   four active compounds which are type of phlorotannins in E. cava and that’s   are dieckol (DK), 2,7-phloroglucinol-6,6-bieckol (PHB),   pyrogallol-phloroglucinol-6,6-bieckol (PPB) and phlorofucofuroeckol A (PFFA)   [16]. (page 2, line 91)

Four   active phlorotannins have been identified in E. cava, namely, dieckol (DK),   2,7-phloroglucinol-6,6-bieckol (PHB), pyrogallol-phloroglucinol-6,6-bieckol   (PPB), and phlorofucofuroeckol A (PFFA) [16].

2.        The present study, we investigated   whether orally administered PPB is an effective treatment for vascular   dysfunction in a high fat diet model of obesity and in a high cholesterol   diet model of hypertension. (page 3, line 98)

The present study was therefore undertaken to   examine the effects of PPB on vascular dysfunction in obese and hypertension   mouse models.

3.        In order to identify the active   phlorotannin of the ECE, we compared the four phlorotannins isolated   from ECE, as previously described [17]. (page 4, line 134)

In order to identify the active component   of the ECE, four phlorotannins isolated from ECE were compared, as previously   described [17].

4.        To   investigate the protective effects of adhesion molecule expression on   SVEC4-10 as mouse endothelial cells, we treated with PBS, 0.25 mM of   PA, or 0.25 mM of PA plus four phlorotannins for 48 hrs on ECs (Figure 2A).   (page 4, 137)

To   investigate the protective effects of adhesion molecule expression on ECs as   mouse endothelial cells, ECs were exposed to either PBS, 0.25 mM of PA, or   0.25 mM of PA plus four phlorotannins for 48 hrs (Figure 2A).

5.      Having found PPB was more active   than the other three active compounds in vitro, we examined it effects   in diet-induced mouse models of obesity and hypertension. (page 6, line 184)

Considering that PPB was more active than the other   three active compounds in vitro, we examined the effects of PPB in   diet-induced mouse models of obesity and hypertension.

6.      In addition, we also   investigated the reduction effects of PPB on the excessive proliferation and   migration of VSMCs in the two animal models. Expression levels of pERK1/2 and   pAKT in VSMCs were highest in DIO/Saline and DIH/Saline mice groups (Figures   5A-C) among them, suggesting they were involved in the observed reductions in   numbers of proliferative VSMCs and in wall thicknesses. (page 7, line 213)

In addition, the   inhibitory effects of PPB on the excessive proliferation and migration of   VSMCs were validated in the two animal models. Expression levels of pERK1/2   and pAKT in VSMCs were highest in the Obesity/Saline and Hypertension/Saline   mice groups (Figures 6)

Point 3: The plants Latin scientific names must be in Italic manner.

Response 3: We appreciate this comment. We checked plant names (Ecklonia cava and Ginkgo biloba) and the names was Italicized throughout the manuscript. Changed examples are summarized below.

1.            Several studies have been conducted on the   relation between Ecklonia cava and cardiovascular health [13-16, 18].

Several studies have been conducted on the   association between Ecklonia cava   and cardiovascular health [13-16, 18].

2.            These   days, people are interested in functional foods or food ingredients like   green tea, nuts, cayenne, Ginkgo biloba (GB), and Ecklonia cava   (E. cava) as a means of improving blood circulation.

Recent times have seen a considerable interest in   the use of functional foods to improve blood circulation, such as green tea,   nuts, cayenne, Ginkgo biloba (GB),   and Ecklonia cava (E. cava).

Point 4: Too many abbreviations throughout the whole manuscript try to reduce them to be this manuscript more readable for other none medical professions.

Response 4: We appreciate this comment. We have checked the abbreviations throughout the manuscript and attempted to reduce them. Typically, diet-induced obesity (DIO) is marked ‘Obese’ and diet-induced hypertension (DIH) is marked ‘Hypertension’. In addition, terms that are infrequently used have not been abbreviated. We have also added an abbreviations section in the manuscript to reduce the confusion of readers.

Abbreviation: palmitate conjugation to BSA; DK, dieckol; PHB,   2,7-phloroglucinol-6,6-bieckol; PPB, pyrogallol-phloroglucinol-6,6-bieckol;   PFFA, phlorofucofuroeckol A; α-SMA, α-smooth muscle actin; PCNA,   Proliferating cell nuclear antigen; H&E, Hematoxylin and eosin.

Point 5: Reference number 2 and many other references are too old and try to find other updated ones.

Response 5: We appreciate this comment. In reference section, some of the references selected by the reviewer have been searched and changed. All 12 references in the Reference section of the manuscript have been replaced with the latest references since 2008, page 11 in below.

Old reference 1: Furchgott, R.F.; Vanhoutte, P.M.   Endothelium-derived relaxing and contracting factors. FASEB J. 1989, 3,   2007-2018.

New reference   1: Vanhoutte, P.M.; Zhao, Y.;   Xu, A.; Leung, S. W. Thirty years of saying NO: sources, fate, actions, and   misfortunes of the endothelium-derived vasodilator mediator. Circulation res.  2016, 2, 375-396.

Old reference 2: Ross,   R.; Glomset, J.A. Atherosclerosis and the arterial smooth muscle cell:   Proliferation of smooth muscle is a key event in the genesis of the lesions   of atherosclerosis. Science 1973, 180, 1332-1329.

New   reference 2: Bennett,   M.R.; Sinha, S.; Owens, G.K. Vascular smooth muscle cells in atherosclerosis.   Circulation res. 2016, 4, 692-702.

Old reference 3: Simoni,   J.; Simoni, G.; Lox, C.D.; Prien, S.D.; Tran, R.; Shires, G.T. Expression of   adhesion molecules and von Willebrand factor in human coronary artery   endothelial cells incubated with differently modified hemoglobin solutions.   Artif. Cells Blood Substit. Immobil.   Biotechnol. 1997, 25, 211-225.  

New   reference 3: Iba, T.;   Levy, J.H. Inflammation and thrombosis: roles of neutrophils, platelets and   endothelial cells and their interactions in thrombus formation during   sepsis. J Thromb Haemost. 2018, 2, 231-241.

Old reference 4: Kleijnen,   J.; Knipschild, P. Ginkgo biloba. Lancet   1992, 340, 1136-1339.

New   reference 4: DeKosky, S.T.; Williamson, J.D.; Fitzpatrick,   A.L.; Kronmal, R.A.; Ives, D.G.; Saxton, J.A.; Lopez, O.L.; Burke, G.;   Carlson, M.C.; Fried, L.P.; Kuller, L.H.; Robbins, J.A.; Tracy, R.P.;   Woolard, N.F.; Dunn, L.; Snitz, B.E.; Nahin, R.L.; Furberg, C.D. Ginkgo   biloba for prevention of dementia: a randomized controlled trial. JAMA. 2008, 19, 2253-2262.

Old reference 5: Mouren,   X.; Caillard, P.; Schwartz, F. Study of the antiischemic action of EGb 761 in   the treatment of peripheral arterial occlusive disease by TcPo2   determination. Angiology 1994, 45, 413-417.

New   reference 5: Nash, K.M.; Shah, Z.A. Current perspectives on the beneficial role of   Ginkgo biloba in neurological and cerebrovascular disorders. Integr Med Insights. 2015, 10, IMI-S25054.

Old reference 6: Krieglstein,   J.; Beck, T.; Seibert, A. Influence of an extract of Ginkgo biloba on   cerebral blood flow and metabolism. Life   Sci. 1986, 39, 2327-2334.

New   reference 6: Zuo, W.; Yan, F.; Zhang, B.; Li, J.; Mei, D. Advances in the studies   of Ginkgo biloba leaves extract on aging-related diseases. Aging Dis2017, 6, 812.

Old reference 7: Stücker,   O.; Pons, C.; Duverger, J.P.; Drieu, K.; D’arbigny, P. Effect of Ginkgo   biloba extract (EGb 761) on the vasospastic response of mouse cutaneous   arterioles to platelet activation. Int.   J. Microcirc. Clin. Exp. 1997,   17, 61-66.

New   reference 7: Tian,   J.; Liu, Y.; Chen, K. Ginkgo biloba extract in vascular protection: molecular   mechanisms and clinical applications. Curr   Vasc Pharmacol. 2017, 15, 532-48.

Old reference 8: Smith,   J.V.; Luo, Y. Studies on molecular mechanisms of Ginkgo biloba extract. Appl. Microbiol. Biotechnol. 2004, 64, 465-472.

New   reference 8: Yang,   Y.; Li, Y.; Wang, J.; Sun, K.; Tao, W.; Wang, Z.; Xiao, W.; Pan, Y.; Zhang,   S.; Wang, Y. Systematic Investigation of Ginkgo Biloba Leaves for Treating   Cardio-cerebrovascular Diseases in an Animal Model. ACS Chem Biol. 2017, 5, 1363-1372.

Old reference 9: Athukorala,   Y.; Jung, W.K.; Vasanthan, T.; Jeon, Y.J. An anticoagulative polysaccharide   from an enzymatic hydrolysate of Ecklonia cava. Carbohydr. Polym. 2006,   66, 184-191.

New   reference 9: Ana,   M.L.; Seca, I.D.; Diana, C.G.A.; Pinto. Overview on the Antihypertensive and Anti-Obesity   Effects of Secondary Metabolites from Seaweeds, Mar Drugs. 2018, 7, E237.

Old reference 10: Hong,   J.H.; Son, B.S.; Kim, B.K.; Chee, H.Y.; Song, K.S.; Lee, B.H.; Shin, H.C.;   Lee, K.B. Antihypertensive effect of Ecklonia cava extract. Kor. J. Pharmacogn. 2006, 37, 200-205.

New   reference 10: Seca, A.; Pinto, D. Overview on the antihypertensive and anti-obesity   effects of secondary metabolites from seaweeds. Marine drugs2018, 7, 237.

Old reference 11: Athukorala,   Y.; Kim, K.N.; Jeon, Y.J. Antiproliferative and antioxidant properties of an   enzymatic hydrolysate from brown alga, Ecklonia cava. Food Chem. Toxicol. 2006,   44, 1065-1074.

New   reference 11: Venkatesan, J.; Kim, S.K.; Shim, M. Antimicrobial, antioxidant, and   anticancer activities of biosynthesized silver nanoparticles using marine   algae Ecklonia cava. Nanomaterials2016, 12, 235.

Old reference 12: DeFeudis,   F.V. A brief history of EGb 761 and its therapeutic uses. Pharmacopsychiatry 2003, 36, Suppl 1:S2-7.

New   reference 12: Beck, S.M.; Ruge, H.; Schindler, C.; Burkart, M.; Miller, R.;   Kirschbaum, C.;  Goschke, T. Effects of   Ginkgo biloba extract EGb 761® on cognitive control functions, mental   activity of the prefrontal cortex and stress reactivity in elderly adults   with subjective memory impairment–a randomized doubleblind placebocontrolled trial. Human   Psychopharmacology: Clinical and Experimental2016, 3, 227-242.

Point 6: Many sentences throughout the manuscript needs to be rephrased.

Response 6: We appreciate this comment. Some sentences selected by the reviewer have been changed in the manuscript, and some sentences have been modified, added, or deleted during the repositioning of figures.

Original 1: First compared the ECE with the GBE to know if the ECE is as   efficacious as the well-known GBE in diet induced obese mice model. next, we   isolated single phlorotannins of ECE using CPC and HPLC and confirmed   inhibitory effects of endothelial cell and vascular smooth muscle cell   dysfunction. (page 1, line 24)

Modification 1: In this study, the beneficial effect of ECE and leaf of Ginkgo biloba   extract (GBE), a well-known dietary supplement, were first confirmed in a   diet induced-obese model. Next, 4 phlorotannins were isolated from ECE, and   their inhibitory effects on vascular cell dysfunction were validated.

Original   2: The present study, we   investigated whether orally administered PPB is an effective treatment for   vascular dysfunction in a high fat diet model of obesity and in a high   cholesterol diet model of hypertension. (page 3, line 98)

Modification   2: The present study was therefore undertaken to examine the effects of   PPB on vascular dysfunction in obese and hypertension mouse models.

Original   3: Figure 1. Comparative   analysis of GBE and ECE on the improvement of vascular dysfunction in vitro   and in vivo (A) PA/PBS, PA/ECE or PA/GBE to SVEC4-10 (mouse endothelial cells)   and survival ratio was measured by cell survival assay. (B) Confocal   microscopic images showing TUNEL positive cells (red; arrows) and DAPI   stained nuclei (blue). Scale bar = 50 μm (c, d) PA/PBS, PA/ECE or PA/GBE to   MOVAS and proliferating and migrating VSMCs ratio were measured by each   assay. (page 3, line 15)

Comment 3: The above sentence was deleted because the figure composition was made again.

Original   4: Figure 4. Inhibitory   effects of the PPB isolated from ECE on EC dysfunction in diet-induced mouse   models of obesity and hypertension (A-E) Arrows in confocal microscopic   images show adhesion molecule expressions (green, arrow; E-selectin, ICAM-1,   VCAM-1, VWF) or nuclei (DAPI, blue) in DIO and DIH mouse models. (page 7,   line 199)

Comment 4: The above sentence was deleted because the figure composition was made again.

Original   5: Figure 5. Inhibitory   effects of the PPB isolated from ECE on VSMC dysfunction in diet-induced   mouse models of obesity and hypertension (A-C) Light microscopic images   showing abnormally high expressions of proliferation and migration related   molecules (pERK1/2 and pAKT) in VSMCs in DIO and in DIH. Quantitative graphs   show intensity from representative images (red bar; DIO mouse model, blue   bar; DIO mouse model). Scale bar = 10 μm (D-G) Light microscopic images   showing proliferation marker (PCNA) and H&E stained blood vessels. Plots   showing numbers of PCNA positive cells and intima-media ratios obtained using   representative H&E images (red bar; DIO mouse model, blue bar; DIO mouse   model). (page 7, line 225)

Comment 5: The above sentence was deleted because the figure composition was made again.

Point 7: Figure 1 4 and 5, need to be divided into two figures (too many images in one figure).

Response 7: We understand the reviewer’s point of view. As mentioned, Figures 1, 4, and 5 each need to be divided into two figures. We have accordingly changed the structure of all figures to help better and easier understanding for our readers. The final manuscript now contains 7 main figures, 3 supplementary figures and 3 tables. Figure legends and results have accordingly been revised in the manuscript.

Please, Look at the attached file.

Point 8: Very short conclusion? you can add some recommendations

Response 8: We appreciate this comment. As per the reviewer’s suggestion, some recommendations have been added to the conclusion section of the manuscript. Page below:

Original version (page 12, line 438)

PPB was better   at protecting ECs from death and at preventing excessive VSMC proliferation   and migration than the other three active compounds isolated from ECE.   Furthermore, PPB as a dietary supplementation was found to reduce blood   pressure and serum lipoprotein levels in these two mouse models.

Modified version

The efficacy of   ECE for improving blood circulation was confirmed by administration to a high   fat diet-induced mouse model for 4 weeks. ECE contains phlorotannins such as   dieckol (DK), 2,7-phloroglucinol-6,6-bieckol (PHB), pyrogallol-phloroglucinol-6,6-bieckol   (PPB) and phlorofucofuroeckol A (PFFA). Especially, PPB markedly reduces   adhesion molecule expression, EC death and excessive migration, and   proliferation of VSMCs in vitro and in the obesity and hypertension mouse   models. Additionally, PPB as a dietary supplement remarkably reduces the   blood pressure and serum lipoprotein levels in vivo. As expected, PPB plays   an important role as the active substance in ECE for improving blood   circulation. Currently, it is anticipated that functional foods, such as   substances that are effective for alleviating various blood circulation   related problems, will be effective in improving blood circulation. It is   believed that such substances have the potential for future applications in   the clinical setting.

Point 9: Many correction required to be established (grammatical and editing) and needs to be checked by native speaker or by any editing proof services

Response 9: We appreciate this comment. As commented by the reviewer, grammatical editing by a native speaker has been performed. Related certificates are attached below.

 Please, Look at the attached file.

Reviewer 2 Report

The main concern about this work by Son et al. is that numerous sections (especially in the methods and results) are extremely confused, and it's hard for the reader to follow how the experiments have been performed and the significance of the results. Due to the relevance of the topic, i.e. the identification of functional food supplements useful for the reduction of cardiovascular risk, I suggest the authors to completely reconsider the paper and organize it better.

Animal model: the mouse model should be better described and justified. Why using only 45% kcal from fat in the diet? Please explain the different models using a Table, not in the text.

In vitro esperiments: why not using primary cells obtained by mice to confirm the data?

The authors should first describe and discuss the in vitro experiments and then the in vivo findings. 

Fig.2F: western blot. the last 3 proteins (caspase 3 and 8 and beta-actin) seem to be taken from a whole membrane, whereas the others seem to be obtained by a cleaved one. why?

Author Response

Response to Reviewer 2 Comments

 Point 1: The main concern about this work by Son et al. is that numerous sections (especially in the methods and results) are extremely confused, and it's hard for the reader to follow how the experiments have been performed and the significance of the results. Due to the relevance of the topic, i.e. the identification of functional food supplements useful for the reduction of cardiovascular risk, I suggest the authors to completely reconsider the paper and organize it better.

  Response 1: We deeply appreciate the reviewer’s point of view. As mentioned, all figures have        been changed and restructured to help our readers understand it easily. The final manuscript   contains 7 main figures, 3 supplementary figures and 3 tables. Figure legends and results have     also been changed appropriately in the manuscript.

Please, Look at the attached file.

Point 2: Animal model: the mouse model should be better described and justified. Why using only 45% kcal from fat in the diet? Please explain the different models using a Table, not in the text.

Response 2: We appreciate and agree with comment. In order to describe and justify these results, we need to explain the different models using a table and figure, which have now been added in supplementary materials as Table S2 and Figure S3, and is accordingly mentioned in the main manuscript (page 9).

Please, look at the attached file.

Point 3: In vitro experiments: why not using primary cells obtained by mice to confirm the data?.

Response 3: We appreciate and agree with the comment. Experiments using primary cells are more relevant than cell lines. However, the purpose of cell experiments (in vitro) in this study was to facilitate the screening of the inhibitory efficacy for EC and VSMC dysfunction of the four compounds (DK, PHB, PPB and PFFA) isolated from Ecklonia cava extract, prior to animal experiments (in vivo). PPB was selected from among the four isolated compounds. The inhibitory efficacy of PPB was then confirmed in the obese- and hypertension- induced mouse models.

Point 4: The authors should first describe and discuss the in vitro experiments and then the in vivo findings.

Response 4: We understand and are in agreement with this comment. The order of in vitro and in vivo experiment findings has accordingly been changed. The in vitro findings are explained first and in vivo findings explained later to help readers understand. The revised section can be found through pages 3-7.

Point 5: Fig.2F: western blot. the last 3 proteins (caspase 3 and 8 and beta-actin) seem to be taken from a whole membrane, whereas the others seem to be obtained by a cleaved one. why?.

Response 5: We appreciate this comment. Our results are extracted from the following experimental results. The results seem to be truncated because the membrane background is darker than the other three proteins (caspase 3 and 8 and beta-actin). In this paper, it is thought that it is difficult for readers to understand because it includes many contents such as pathway analysis, including AMPK. Therefore, to facilitate the reader's understanding, we reorganized the main figure and results. Accordingly, Western blotting results have been removed in Figure 2 and Figure 3, and we have concentrated on EC death, VSMC proliferation and migration in obese- and hypertension-induced mouse models. The revised figures can be found through page 4.

Please, Look at the attached file.

Round  2

Reviewer 1 Report

I have no more comments

Author Response

 Thank you very much for your comments.

Reviewer 2 Report

The paper has been significantly improved. However, ad additional editing of the English language should be performed. The conclusions should be more effective. 

Author Response

Please see the attached letter.
